# Dose-Dependent Solubility–Permeability Interplay for Poorly Soluble Drugs under Non-Sink Conditions

**DOI:** 10.3390/pharmaceutics13030323

**Published:** 2021-03-02

**Authors:** Kazuya Sugita, Noriyuki Takata, Etsuo Yonemochi

**Affiliations:** 1Department of Physical Chemistry, Hoshi University, 2-4-41, Ebara, Shinagawa, Tokyo 142-8501, Japan; sugita.kazuya55@chugai-pharm.co.jp; 2Quality Development Department, Chugai Pharma Manufacturing Co., Ltd., 5-5-1, Ukima, Kita, Tokyo 115-8543, Japan; takatanry@chugai-pharm.co.jp

**Keywords:** non-sink condition, solubility–permeability interplay, unstirred water layer, poorly soluble drugs, solubilizer additive

## Abstract

We investigated the solubility–permeability interplay using a solubilizer additive under non-sink conditions. Sodium lauryl sulfate (SLS) was used as a solubilizer additive. The solubility and permeability of two poorly soluble drugs at various doses, with or without SLS, were evaluated by flux measurements. The total permeated amount of griseofulvin, which has high permeability, increased by the addition of SLS. On the other hand, triamcinolone, which has low permeability, showed an almost constant rate of permeation regardless of the SLS addition. The total permeated amount of griseofulvin increased by about 20–30% when the dose amount exceeded its solubility, whereas its concentration in the donor chamber remained almost constant. However, the total permeated amount of triamcinolone was almost constant regardless of dose amount. These results suggest that the permeability of the unstirred water layer (UWL) may be affected by SLS and solid drugs for high-permeable drugs. The effect of solid drugs could be explained by a reduction in the apparent UWL thickness. For the appropriate evaluation of absorption, it would be essential to consider these effects.

## 1. Introduction

In vitro tools that can assess the absorption performance of solid oral formulations, such as a dissolution test, play an important role in pharmaceutical development [1,2,3,4,5,6,7,8]. These in vitro tools have multiple purposes: selecting the formulation in the pre-clinical stage [2,3,4,5], optimizing both the formulation and the manufacturing process in the clinical stage [5], and conducting quality control and bioequivalence studies in the commercial stage [9,10,11,12,13,14]. Recent studies show that more than 70% of drug candidates have low solubility, classified in the Biopharmaceutical Classification System (BCS) as BCS class II or IV [3,4,15]. Their low solubility can dramatically limit their absorption. To increase the solubility of these drugs, various solubilizer additives, like surfactants and cyclodextrins (CDs), are often added to the formulations [16,17]. 

The most commonly used in vitro tool, dissolution testing, measures the dissolution rate and solubility of drugs to assess their absorption performance. However, some studies report that the results from the in vitro dissolution testing of formulations that include solubilizers often fail to predict the in vivo absorption [6,18,19,20,21,22]. A major reason for these inconsistent results may be found in the solubility–permeability interplay, wherein solubilizer additives increase drug solubility but decrease permeability. Some papers have reported that even though solubilizer additives successfully increase drug solubility, this interplay hinders in vivo absorption [23,24,25,26].

Intestinal membrane transport of drugs in the gastrointestinal tract (GIT) involves two processes: transcellular diffusion and paracellular diffusion [27,28] (pp. 297–307). Transcellular diffusion is generally considered to determine drug permeability. Transcellular diffusion can be further divided into two processes: passive diffusion and active transport by carriers/transporters. Both processes affect the permeability of drugs, but the solubility–permeability interplay only occurs during passive diffusion; therefore, this study will focus only on passive diffusion. 

Drug molecules dissolved in the GIT after oral administration can take on a variety of new forms; they can be ionized, captured in micelles, or drawn into complexes with other molecules. Among these, the un-ionized free molecules—those not tethered to complexes—are the ones that mainly affect permeability, and they are called un-ionized free drugs (UFDs) [29,30]. Solubilizer additives like surfactants and CDs increase the apparent solubility of drugs by forming micelles and trapping molecules within them or by directly binding to the molecules. However, solubilizers do not change the amount of UFDs, and therefore, the fraction of UFDs in the dissolved molecules decreases, which means the apparent permeability of drugs also decreases because of the solubilizers. The effect of the solubility–permeability interplay on drug absorption is a big issue that has been investigated in both in vitro and in vivo studies. Beig et al. examined the solubility–permeability interplay of CDs using in vitro studies [31,32,33]. Miller et al. confirmed a similar interplay effect by sodium lauryl sulfate (SLS) also using in vitro studies [34]. Hens et al. studied the effect of bile micelles on the solubility–permeability interplay by measuring the in vivo absorption of fenofibrate in healthy volunteers [35]. 

A lot of BCS class II and IV drugs cannot be dissolved completely in the gastrointestinal tract (GIT). When a drug is not completely dissolved, the condition is referred to a non-sink condition. Therefore, to appropriately evaluate drug absorption, we must understand the solubility–permeability relationship under realistic non-sink conditions. Unfortunately, most studies on the solubility–permeability interplay involve sink conditions in which drugs are dissolved completely. Under sink conditions, the effect of the solubility–permeability interplay on drug absorption can be explained by a mechanism involving two continuous processes in passive diffusion; the first is the diffusion of drug molecules in the unstirred water layer (UWL) on the membrane surface, and the second is diffusion in the membrane itself. As the membrane is composed of phospholipids, drug molecules should be in lipophilic form to be partitioned to the membrane. Because a UFD is far more lipophilic in the GIT than in any other form, it is thought that a UFD alone can permeate during passive diffusion. This theory has been confirmed by numerous studies over the years [30]. Therefore, it is assumed that the UFD amount alone determines permeability in the membrane. However, in the UWL, drug molecule diffusion could be affected by other forms, in addition to the UFD [36]. UWL diffusion may depend on the unique properties of drug molecules and solubilizer additives. Some studies have successfully confirmed the solubility–permeability interplay by solubilizer additives under sink conditions using the absorption mechanism described above [31,32,33,34]. However, it has also been reported that undissolved solid drugs could affect the diffusion of drug molecules in the UWL under non-sink conditions [37,38,39]. These studies suggest that the permeation of some drugs under non-sink conditions might be faster than that under sink conditions. Unfortunately, there have been too few studies on the solubility–permeability interplay under non-sink conditions.

The purpose of this study is to investigate the relationship between drug solubility, the amount of UFDs, and permeability under non-sink conditions. We used SLS, which is often used as a solubilizer additive in the formulations, and we used griseofulvin and triamcinolone, which have very different solubilities and lipophilicities, as model compounds (Figure 1 and Table 1). We measured the solubility and permeability of these compounds with or without SLS. To investigate the difference between sink and non-sink conditions, the sample dose was changed for each measurement. 

This study can clarify the effect of solubilizer additives and sample dosage on drug permeability and absorption under non-sink conditions.

## 2. Theoretical Basis

In discussing how the solubility–permeability interplay is mediated by solubilizer additives, such as surfactants, we assume that the drug permeability is determined by passive diffusion and that drug molecules are not ionized in the solvent. When solubilizer additives increase apparent solubility, the theoretical relationship between solubility and permeability is described as follows.

Based on Fick’s first law, flux in the drug permeation can be expressed as shown in Equation (1): [36,43]
(1)$$J\;\left(t\right)={\;P}_{\mathrm{app}}C_{\mathrm{app}}\left(t\right)$$
where P_app_ is the effective or apparent permeability and C_app_(t) is the apparent drug concentration.

As described in the introduction, overall permeation can be divided into membrane permeation and UWL permeation. Thus, the apparent permeability (P_app_) is calculated using both apparent membrane permeability (P_m (app)_) and apparent UWL permeability (P_UWL (app)_), as shown in Equation (2) [43,44]:(2)$$\frac{1}{P_{\mathrm{app}}}=\;\frac{1}{P_{m\;\left(\mathrm{app}\right)}}\;+\;\frac{1}{P_{\mathrm{UWL}\;\left(\mathrm{app}\right)}}$$

P_m (app)_ is calculated using the fraction of UFDs (F_U_) as shown in Equation (3) [45]:(3)$$P_{m\;\left(\mathrm{app}\right)}={\;F}_{U}P_{m\;\left(U\right)}$$
where P_m (U)_ is the intrinsic permeability of UFDs. F_U_ is calculated by the solubility of drugs as shown in Equation (4):(4)$$F_{U}=\;\frac{S_{U}}{S}$$
where S_U_ is the intrinsic solubility of drugs in the absence of solubilizer additives and S is the apparent solubility of drugs. 

P_UWL (app)_ can be expressed using the apparent aqueous diffusivity (D_aq (app)_) from Fick’s first law as shown in Equation (5): (5)$$P_{\mathrm{UWL}\;\left(\mathrm{app}\right)}=\;\frac{D_{\mathrm{aq}\;\left(\mathrm{app}\right)}}{h_{\mathrm{UWL}\;\left(\mathrm{app}\right)}}$$
where h_UWL (app)_ is the apparent thickness of the UWL. When there are no solubilizer additives in the drug formulation or test media in the permeability measurements, Equation (5) can be described as shown in Equation (6):(6)$$P_{\mathrm{UWL}\;\left(U\right)}=\;\frac{D_{\mathrm{aq}\;\left(U\right)}}{h_{\mathrm{UWL}\;\left(U\right)}}$$
where h_UWL (U)_ is the intrinsic thickness of the UWL and D_aq (U)_ is the intrinsic aqueous diffusivity of UFDs. The thickness of the UWL can be determined based on rotation speed in the flux experiment [46,47]. If the rotation speed is constant, we can assume that the thickness of the UWL is constant, independent of the addition of solubilizer additives, as shown in Equation (7):(7)$$h_{\mathrm{UWL}\;\left(\mathrm{app}\right)}={\;h}_{\mathrm{UWL}\;\left(U\right)}$$

Substituting h_UWL (app)_ of Equation (5) and h_UWL (U)_ of Equation (6) for Equation (7), P_UWL (app)_ is described as shown in Equation (8):(8)$$P_{\mathrm{UWL}\;\left(\mathrm{app}\right)}={\;P}_{\mathrm{UWL}\;\left(U\right)}\times \frac{D_{\mathrm{aq}\;\left(\mathrm{app}\right)}}{D_{\mathrm{aq}\;\left(U\right)}}$$

The apparent aqueous diffusivity (D_aq (app)_) is a combined total of each fraction of aqueous diffusivity as shown in Equation (9) [36]:(9)$$D_{\mathrm{aq}\;\left(\mathrm{app}\right)}={\;F}_{U}D_{\mathrm{aq}\;\left(U\right)}\;+\;\left(1-F_{U}\right)D_{\mathrm{aq}\;\left(B\right)}$$
where F_U_ is the same fraction of UFDs as in Equation (4) and D_aq (B)_ is the aqueous diffusivity of drug molecules bound to solubilizer additives. P_UWL (app)_ can be calculated by Equations (6), (8), and (9) as shown in Equation (10):(10)$$P_{\mathrm{UWL}\;\left(\mathrm{app}\right)}=\;\frac{1}{h_{\mathrm{UWL}\;\left(\mathrm{app}\right)}}\;\left\{F_{U}D_{\mathrm{aq}\left(U\right)}\;+\;\left(1-F_{U}\right)D_{\mathrm{aq}\;\left(B\right)}\right\}$$

If we know the aqueous diffusivity of each component and the thickness of the UWL, the apparent permeability P_app_ can be calculated based on the drug solubility using Equations (2), (3), and (10).

## 3. Materials and Methods

### 3.1. Materials

Griseofulvin and triamcinolone were purchased from Tokyo Chemical Industry Co., Ltd. (Tokyo, Japan). Buffer components (sodium dihydrogen phosphate (NaH_2_PO_4_), sodium hydroxide (NaOH), sodium chloride (NaCl)), acetonitrile (MeCN), trifluoroacetic acid (TFA), ethylene glycol, N,N-dimethylacetamide (DMA), and dimethyl sulfoxide (DMSO) were purchased from FUJI FILM Wako Pure Chemical Co. (Osaka, Japan). MeCN and TFA were HPLC grade. Ethylene glycol and DMA were Wako special grade. DMSO was a guaranteed reagent. The gastrointestinal tract (GIT) lipid and the acceptor sink buffer (ASB) were purchased from Pion Inc. (Billerica, MA, USA).

### 3.2. Methods

#### 3.2.1. Flux Measurements

The flux was measured by MicroFlux^TM^ (Pion Inc., Billerica, MA, USA). Two chambers in this device were separated by a polyvinylidene fluoride (PVDF) membrane filter of 0.45 μm pore size with 25 μL of GIT lipid solution (Pion Inc., Billerica, MA, USA). Next, 20 mL of ASB (Pion Inc., Billerica, MA, USA) was added into each acceptor chamber, and 20 mL of the test sample solution was added to each donor chamber. Details of the test sample solution in the donor chambers are summarized in Table 2. The sample dose amount was set based on the preliminary solubility study. Then, 5 μg/mL of griseofulvin and 100 μg/mL of triamcinolone were evaluated under sink conditions, and other samples were evaluated under non-sink conditions. The SLS concentration was set based on the reported SLS amount used in the drug formulation and the reported solvent volume in human GIT. Japan’s Ministry of Health, Labor and Welfare reported that the maximum SLS amount administered was 300 mg [48]. The solvent volume of the small intestine in a fasted condition, where most of orally administered drug is absorbed, was reported to be around 50–300 mL [49,50]. Therefore, the maximum SLS concentration in the human small intestine could be around 0.10–0.60% (*w/w*). If enough SLS was used in the drug formulation as solubilizer additive, more than 0.05% (*w/w*) SLS could be included in the solvent of the small intestine. To prepare 5 μg/mL of griseofulvin and 100 μg/mL of triamcinolone, 19.9 mL of test media and 0.1 mL of the concentrated sample solution prepared by DMSO were mixed. To prepare other test sample solutions, the determined test sample amount was added to a test tube containing test media. These samples were placed on the rotation stirrers in a water bath at 37 °C and stirred well. After stirring, these samples were suspended visually. The measurements were started by adding 20 mL of these test sample solutions to the donor chambers. Cross-bar magnetic stirrers were located in each chamber, rotating at 150 rpm. The media in the donor and the acceptor chambers were maintained at 37 °C during measurement. All flux measurements were performed in triplicate.

The concentration–time profiles were determined via manual sampling from the donor chambers and acceptor chambers during flux measurements. At 0, 30, 60, 120, 240, and 360 min, 100 μL of the acceptor chamber solution was withdrawn and diluted to 100 μL (2× dilution) of 3:2 DMA:ethylene glycol (*v/v*). At the same time point, 400 μL of the donor chamber solution was withdrawn and filtered through a polytetrafluoroethylene (PTFE) membrane filter of 0.22 μm pose size. Then, 100 μL of the filtered solution was diluted to 100 μL (2× dilution) of 3:2 DMA:ethylene glycol (*v/v*). The sample concentration was determined by ultra-high-performance liquid chromatography (UHPLC).

The solubility of each model compound at 37 °C for each test medium was determined by using the donor chamber sample concentration at 0 min of 1000 μg/mL dose for griseofulvin and 10,000 μg/mL dose for triamcinolone.

From the obtained concentration–time profiles in the acceptor chambers, the flux (J) was calculated. The flux refers to the mass transfer through the membrane, and it is defined as the total amount of material crossing one unit area of the membrane per unit time, as described by Equation (11):(11)$$J\;\left(t\right)=\frac{1}{A}\cdot \frac{\mathrm{dm}}{\mathrm{dt}}=\;\frac{V}{A}\cdot \frac{\mathrm{dC}\left(t\right)}{\mathrm{dt}}$$
where dm/dt (μg/mL) is the total amount of material crossing the membrane per unit time, *A* is the area of the membrane (1.54 cm^2^), V is the volume of the acceptor chamber (20 mL), and dC(t)/dt (μg/(mL⋅min)) is the slope of the concentration–time profiles in the acceptor chambers. Time intervals were selected in each test to exclude the lag time of the concentration–time profile and calculate the initial flux. The selected time intervals are described in the Results section. Based on Fick’s first law, assuming the sink condition in the acceptor chambers, the flux can be described by Equation (12):(12)$$J\;\left(t\right)=P_{\mathrm{app}}C_{D}\left(t\right)$$
where P_app_ is the apparent permeability of drugs and C_D_ (t) is the drug concentration in the donor chambers. As the initial flux was calculated in this study, C_D_ (t) at 0 min (=C_D_ (0)) was used to calculate P_app_ by Equation (12).

#### 3.2.2. Sample Concentration Measurements by UHPLC

The sample concentrations were measured on a Waters (Milford, MA) Acquity UPLC H-Class system. An Acquity UPLC^®^ BEH Shield RP18 1.7 μm, 2.1 × 50 mm, was used for chromatographic separation. A gradient mobile phase, spanning 95:5 to 0:100 (*v/v*) water:MeCN (both containing 0.05%TFA) over 2.0 min, was pumped at a flow rate of 1.0 mL/min. The injection volume and ultraviolet (UV) wavelength for griseofulvin were 5 μL and 240 nm, respectively. The injection volume and ultraviolet (UV) wavelength for triamcinolone were 1 μL and 292 nm, respectively.

## 4. Results

### 4.1. Effect of SLS on Griseofulvin Solubility and Triamcinolone Solubility

The solubility of griseofulvin and triamcinolone in each test medium at 37 °C is shown in Table 3. The solubility of griseofulvin increased by about 2.5-fold by the addition of 0.05% SLS. In contrast, the solubility of triamcinolone was mostly constant, independent of the addition of SLS. Compared with triamcinolone, griseofulvin is a lipophilic compound and is easy to solubilize using SLS. 

As both griseofulvin and triamcinolone are neutral compounds, we can assume that these drug molecules are not ionized in the aqueous test media. We can assume that these compounds formed only UFDs in pH 6.5 buffer and that UFDs and the drug molecules bound to SLS in pH 6.5 buffer + 0.05% SLS. Under non-sink conditions, the amount of UFDs would be constant regardless of the SLS amount. Therefore, the amount and fraction of UFDs were estimated by Equation (4) using the average solubility (Table 3).

### 4.2. Effect of Dose Amount and SLS on Flux and Permeability

#### 4.2.1. Griseofulvin

The concentration–time profiles of griseofulvin in the acceptor chamber as determined by the flux measurements are shown in Figure 2. Those in the donor chamber as determined by the flux measurements are shown in Figure A1. The flux calculated using Equation (11) is shown in Figure 3. 

Based on the solubility in Table 3, as for both pH 6.5 buffer and pH 6.5 buffer + 0.05% SLS, the 5 μg/mL dose samples in the donor chamber were under sink conditions and the 50, 200, and 1000 μg/mL dose samples were under non-sink conditions. The sample concentration in the donor chambers for the 5 μg/mL dose correlated well with the prepared sample concentration, and the test samples were visually confirmed to be a transparent solution. On the other hand, the sample concentrations in the donor chamber for the 50, 200, and 1000 μg/mL doses was about 10 μg/mL in pH 6.5 buffer and about 25 μg/mL in pH 6.5 buffer + 0.05% SLS, which were lower than the prepared sample concentrations. These samples were visually confirmed to be suspensions.

The fluxes under sink conditions were smaller than those under non-sink conditions in both pH 6.5 buffer and pH 6.5 buffer + 0.05% SLS; this is because the sample concentrations in the donor chambers under sink conditions were lower than those under non-sink conditions. Under sink conditions, at the 5 μg/mL dose, the flux in pH 6.5 buffer + 0.05% SLS was almost half that in pH 6.5 buffer, even though the sample concentration at 0 min in the donor chamber showed almost the same value. Under non-sink conditions, at the 50, 200, and 1000 μg/mL doses, the flux in pH 6.5 buffer + 0.05% SLS was slightly higher than that in pH 6.5 buffer. Based on Equation (12), these results suggest that the apparent permeability in pH 6.5 buffer + 0.05% SLS was almost half that in pH 6.5 buffer.

In addition, in both pH 6.5 buffer and pH 6.5 buffer + 0.05% SLS, the flux increased by about 10–30%, depending on the dose amount under non-sink conditions, even though the sample concentration at 0 min in the donor chamber showed almost the same value.

#### 4.2.2. Triamcinolone

The concentration–time profiles of triamcinolone in the acceptor chamber, as determined by the flux measurements, are shown in Figure 4, while those in the donor chamber, as determined by the flux measurements, are shown in Figure A2. The flux calculated using Equation (11) is shown in Figure 5. 

Based on the solubility in Table 3, for both pH 6.5 buffer and pH 6.5 buffer + 0.05% SLS, the 100 μg/mL dose test samples in the donor chamber were under sink conditions and the 500, 2000, and 10,000 μg/mL dose test samples in the donor chamber were under non-sink conditions. The sample concentration in the donor chamber for the 100 μg/mL dose correlated well with the prepared sample concentration, and the test samples were visually confirmed to be a transparent solution. On the other hand, the sample concentrations in the donor chamber for 500, 2000, and 10,000 μg/mL doses showed about 200 μg/mL in pH 6.5 buffer and pH 6.5 buffer + 0.05% SLS, which were lower than the prepared sample concentrations. These samples were visually confirmed to be suspensions.

The fluxes under sink conditions were smaller than those under non-sink conditions in both pH 6.5 buffer and pH 6.5 buffer + 0.05% SLS; this is because the sample concentrations in the donor chambers under sink conditions were lower than those under non-sink conditions. Under sink conditions, at the 100 μg/mL dose, the flux in pH 6.5 buffer + 0.05% SLS and pH 6.5 buffer showed almost similar values. Under non-sink conditions, in pH 6.5 buffer, the flux at all doses was also similar. In pH 6.5 buffer + 0.05% SLS, the flux at 500 and 2000 μg/mL doses was similar. At the 10,000 μg/mL dose, the flux and the donor concentration were about 10% higher than those at the 500 and 2000 μg/mL doses. Based on Equation (12), unlike griseofulvin, in both pH 6.5 buffer and pH 6.5 buffer + 0.05% SLS, the apparent permeability was mostly constant and not dependent on the sample dose amount under non-sink conditions.

### 4.3. Theoretical Calculation about Permeability

In this section, we calculated the permeabilities of compounds theoretically using the results and the physicochemical properties of compounds, also applying the equations in the Theoretical Basis section.

In pH 6.5 buffer under sink conditions, because both griseofulvin and triamcinolone are neutral compounds, all the drug molecules in the donor chamber would be UFDs. The intrinsic permeability of UFDs (P_app (U)_) was calculated by Equation (12) using the drug concentration in the donor chamber and the measured flux (measured P_app (U)_ in Table 4). It is reported that the intrinsic aqueous diffusivity of UFDs (D_aq (U)_) depends on the molecular weight (MW) and can be empirically estimated by Equation (13) [51] (p. 381):(13)$${\mathrm{Log}\;D}_{\mathrm{aq}\;\left(U\right)}=\;-4.131-0.4531\mathrm{Log}\;\mathrm{MW}$$

Using the MW of the model compounds, each D_aq (U)_ was calculated (calculated D_aq (U)_ in Table 4). The UWL thickness (h_UWL (app)_ or h_UWL (U)_) for the MicroFlux^TM^ measurements in this study was estimated to be around 100 μm based on previous studies [43,44,51]. The intrinsic UWL permeability of UFDs (P_UWL (U)_) was then calculated by Equation (6) (calculated P_UWL (U)_ in Table 4). Substituting P_app (U)_ and P_UWL (U)_ in Equation (2), the intrinsic membrane permeability of UFD (P_m (U)_) was calculated (calculated P_m (U)_ in Table 4). The results suggested that the permeability of griseofulvin is affected by not only P_m (U)_ but also P_UWL (U)_, and that of triamcinolone can be determined by only P_m (U)_. With triamcinolone, P_UWL (U)_ is much smaller than P_m (U)_. 

For both model compounds, in pH 6.5 buffer + 0.05% SLS under sink conditions, the drug molecules in the donor chamber were UFDs and bound to SLS. The apparent permeability (P_app_) was calculated by Equation (12) using the drug concentration in the donor chamber and the measured flux (measured P_app_ in Table 5). The apparent membrane permeability (P_m (app)_) was calculated by Equation (3) using the fraction of UFDs (F_U_) in Table 3 and P_m (U)_ (calculated P_m (app)_ in Table 5). Substituting P_app_ and P_m (app)_ in Equation (2), the apparent UWL permeability (P_UWL (app)_) was calculated (calculated P_UWL (app)_ in Table 5). The apparent aqueous diffusivity (D_aq (app)_) was calculated by Equation (5) using P_UWL (app)_ and h_UWL (app)_ = 100 μm (calculated D_aq (app)_ in Table 5). Substituting F_U_, D_aq (app)_, and D_aq (U)_ in Equation (9), the aqueous diffusivity of drug molecules bound to SLS (D_aq (B)_) was calculated (calculated D_aq (B)_ in Table 5). As for triamcinolone, D_aq (B)_ was not calculated in this study. The reason for this is the fraction of the drug molecules bound to SLS in the donor chamber would be too small to calculate the impact of SLS-bound molecules on the apparent permeability using the method described above. As for griseofulvin, assuming that Equation (13) applies to SLS-bound drug molecules, their sizes were estimated to be around MW 20,000 Da. As the aggregated number of SLS in aqueous solutions was reported to be 62, their micellar weight appeared to be around 18,000 Da [52]. Therefore, the calculated D_aq (B)_ agreed roughly with the reported micellar size formed by SLS. Assuming this micellar size is constant, permeability and flux could be calculated by Equation (10) and Equation (12) using the apparent solubility. The apparent solubility increased in proportion to the amount of SLS. When the sample concentration in the donor chamber remained at 5 μg/mL under sink conditions, flux and apparent permeability decreased as apparent solubility increased, as shown in Figure 6. For example, if the apparent solubility increased by 5 and 10 times, the apparent permeability and flux reduced to about 1/4 and 1/7, respectively.

## 5. Discussion

The results of this study suggest that the impact of SLS on permeability and flux under sink conditions is different from that under non-sink conditions. Additionally, we confirmed that permeability and flux could increase depending on the sample dose amount in the donor chamber. In this section, we discuss the solubility–permeability relationship and the absorption mechanism under non-sink conditions. 

Under non-sink conditions, the apparent solubility of griseofulvin increased by the addition of SLS, and flux was calculated using Equations (10) and (12) (Figure 7). The equations express the increase in the drug amount diffusing in the UWL, which was due to the SLS-bound drug molecules. The enhancement of flux by the addition of SLS was consistent with the measured values at each sample dose.

In addition to UFDs and SLS-bound drugs, solid drugs in the donor chamber are also under non-sink conditions. Some studies indicate that solid drugs can exist in the UWL on the membrane [37,53,54,55,56]. A nearly saturated high-concentration layer is known to form on the surface of solid drugs [57] (pp. 263–264). If solid drugs are in the UWL under non-sink conditions, then the drug diffusion mechanism in the UWL may be different under sink and non-sink conditions (Figure 8). Under sink conditions, drug molecules in the UWL would be diffused from the UWL interface to the membrane. Under non-sink conditions, in addition to diffusion in the UWL, the drug molecules could also be diffused from the high-concentration layer on the solid drugs in the UWL to the membrane. This solid drug diffusion could decrease the apparent UWL thickness, as described in Figure 8. Because the drug molecules could be diffused from the surface of solid drug, the apparent UWL thickness reduction could depend on the total surface area of solid drugs in the UWL. If the amount of solid drug increases or the particle size decreases, the apparent UWL thickness could also decrease, increasing permeability and flux as a result. Some studies have proposed a similar absorption mechanism, demonstrating this solid drug effect on in vivo and in vitro absorption [37,38,39].

Based on this absorption mechanism, the apparent UWL thickness in this study decreased as the sample dose increased in the donor chamber. As shown in Figure 2 and Figure 3, griseofulvin permeability and flux in both pH 6.5 buffer and pH 6.5 buffer + 0.05% SLS increased with the sample dose under non-sink conditions. When pH 6.5 buffer is in the donor chamber, the apparent UWL thickness is expressed using Equations (2) and (6) as shown in Equation (14):(14)$$h_{\mathrm{UWL}\;\left(\mathrm{app}\right)}=D_{\mathrm{aq}\;\left(U\right)}\left(\frac{1}{P_{\mathrm{app}}}-\frac{1}{P_{m\;\left(U\right)}}\right)$$

When pH 6.5 buffer + 0.05% SLS is in the donor chamber, the apparent UWL thickness is expressed using Equations (2) and (5) as shown in Equation (15):(15)$$h_{\mathrm{UWL}\;\left(\mathrm{app}\right)}=D_{\mathrm{aq}\;\left(\mathrm{app}\right)}\left(\frac{1}{P_{\mathrm{app}}}-\frac{1}{P_{m\;\left(\mathrm{app}\right)}}\right)$$

As for griseofulvin, substituting measured P_app_, P_m (U)_, P_m (app)_, D_aq (U)_, and D_aq (U)_ in Equations (14) and (15), the apparent UWL thickness for each test condition was calculated, as shown in Figure 9. When the sample dose increased from 50 μg/mL to 1000 μg/mL, the apparent UWL thickness reduced from 100 μm to around 60–70 μm. When the apparent solubility increased by SLS, the effect of UWL thinning on the P_app_ of griseofulvin was calculated using Equations (2), (3), and (9) as shown in Equation (16): (16)$$P_{\mathrm{app}}=\frac{1}{F_{U}P_{m\;\left(U\right)}}-\frac{h_{\mathrm{UWL}\;\left(\mathrm{app}\right)}}{F_{U}D_{\mathrm{aq}\left(U\right)}\;+\;\left(1-F_{U}\right)D_{\mathrm{aq}\;\left(B\right)}}$$

If the test sample in the donor chamber was under non-sink conditions, the sample concentration would be equal to the apparent solubility. Therefore, the effect of UWL thinning on the flux of griseofulvin could be estimated using Equation (1) and Equation (16), as shown in Figure 10. This explains how SLS addition and an increase in dose enhance flux.

As the MW of oral pharmaceutical drugs is generally reported to be around 200–1000 Da, the variability of D_aq (U)_ might be very small, based on Equation (13), compared with the variability of P_m (U)_. P_m (U)_ is generally known to show a large variation. To simply calculate the effect of UWL thinning on the permeability and flux of various drugs, we used a neutral model compound with an MW of 400 Da and pH 6.5 buffer as the donor chamber test media for the following calculations. When the P_m (U)_ of this model compound was 0.001–1 cm/min, the relationship between UWL thickness and flux/permeability was calculated by Equation (17), as shown in Figure 11:(17)$$P_{\mathrm{app}}=\frac{1}{P_{m\;\left(U\right)}}-\frac{h_{\mathrm{UWL}\;\left(\mathrm{app}\right)}}{D_{\mathrm{aq}\;\left(U\right)}}$$

The P_m (U)_ of triamcinolone was around 0.0003 cm/min. According to Figure 11, the permeability and flux for triamcinolone would be constant even if the apparent UWL thickness decreased. Therefore, the results for triamcinolone under non-sink conditions are concordant with the absorption model described above.

In addition, Figure 11 indicates that the permeability and flux of highly permeable drugs can be strongly affected by the presence of solid drugs in the UWL. If the UWL thickness reduces by 50% for a drug with a P_m (U)_ of 1 cm/min, its permeability and flux would increase by about twofold. However, it is difficult to directly predict the in vivo absorption amount using the solubility–permeability relationship estimated by MicroFlux^TM^. Factors like movement, surface area, and solvent volume in the human GIT differ from MicroFlux^TM^ conditions. Thus, the effect of solid drugs on UWL permeability may differ in vitro and in vivo. As for griseofulvin, Sugano suggested that solid drugs in the UWL could increase the in vivo permeability and in vivo drug absorption at a 500 mg dose by about twofold [37]. According to this study, the in vivo effect of solid drugs on permeability and drug absorption is larger than the in vitro effect seen in the flux measurements. The in vitro P_m (U)_ of griseofulvin was 0.0284 cm/min in this study. Thus, it could be roughly predicted that the in vivo permeability and drug absorption of a drug with an in vitro P_m (U)_ = 0.005 cm/min would increase by about 20–30% if the UWL thickness was reduced by half at the higher sample dose.

## 6. Conclusions

In this study, we examined the relationship between drug solubility, the amount of UFDs, and permeability under non-sink conditions. We found that drug molecules bound to solubilizer additives and solid drugs can enhance UWL permeability, resulting in increased flux or permeability under non-sink conditions. Conventional methods of measuring permeability, like Caco-2 or Parallel Artificial Membrane Permeability Assay (PAMPA), are performed under sink conditions. On the other hand, a lot of BCS class II and IV drugs cannot be adequately dissolved in the GIT, meaning that they are under non-sink conditions there. If we use conventional methods to predict the absorption of highly permeable drugs, the predicted amount could be much lower than the actual amount. Therefore, as the results of this study suggest, it is necessary to understand how the solubility–permeability relationship under non-sink conditions reflects the clinical context. MicroFlux^TM^ and the test conditions used in this study may be effective tools for assessing such effects on drug absorption. Not enough is known about the correlation between in vivo permeability and permeability, as estimated by MicroFlux^TM^. However, by combining flux measurements with physiologically based pharmacokinetics (PBPK) modeling, we will be able to accurately predict the in vivo absorption for BCS class II and IV drugs in the future.

## Figures and Tables

**Figure 1 pharmaceutics-13-00323-f001:**
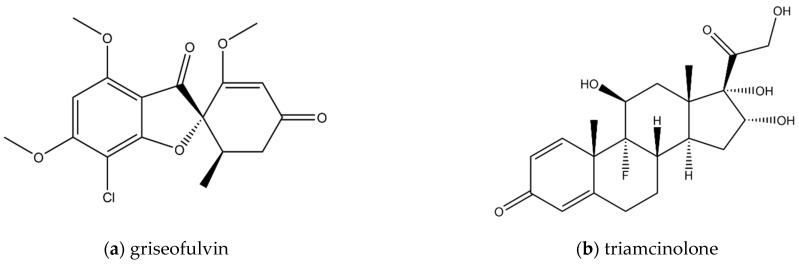
The molecular structure of griseofulvin (**a**) and triamcinolone (**b**).

**Figure 2 pharmaceutics-13-00323-f002:**
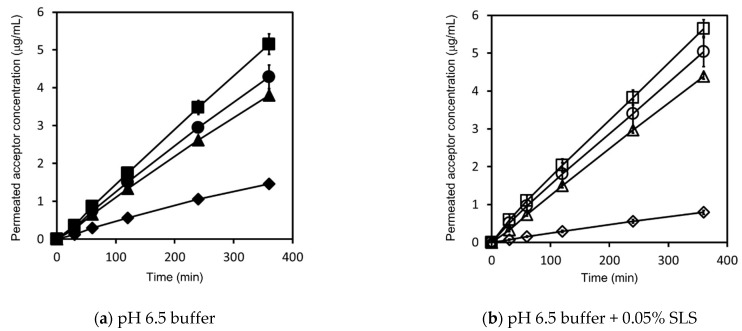
Griseofulvin concentration–time profile in the acceptor chamber as determined by flux measurements. Measurements in pH 6.5 buffer are represented by closed symbols for the 5 μg/mL sample dose (◆), 50 μg/mL sample dose (▲), 200 μg/mL sample dose (●), and 1000 μg/mL sample dose (■) (**a**). Measurements in pH 6.5 buffer + 0.05% Sodium lauryl sulfate (SLS) are represented by open symbols for the 5 μg/mL sample dose (◇), 50 μg/mL sample dose (△), 200 μg/mL sample dose (○), and 1000 μg/mL sample dose (□) (**b**). Results represent the average permeated griseofulvin concentration ± SD (*n* = 3).

**Figure 3 pharmaceutics-13-00323-f003:**
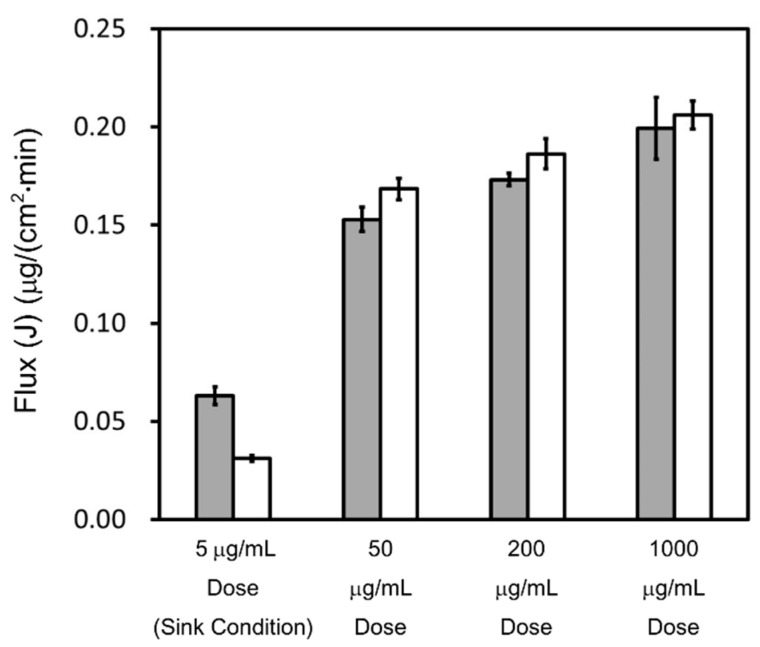
Calculated flux of griseofulvin. A time interval of 30–120 min was selected. The flux in pH 6.5 buffer is represented by gray bars. The flux in pH 6.5 buffer + 0.05% SLS is represented by white bars. Results represent the average griseofulvin flux ± SD (*n* = 3).

**Figure 4 pharmaceutics-13-00323-f004:**
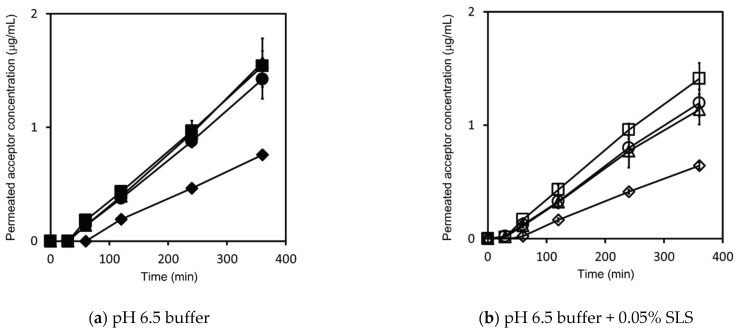
Triamcinolone concentration–time profile in the acceptor chamber, as determined by flux measurements. Measurements in pH 6.5 buffer are represented by closed symbols for the 100 μg/mL sample dose (◆), 500 μg/mL sample dose (▲), 2000 μg/mL sample dose (●), and 10,000 μg/mL sample dose (■) (**a**). Measurements in pH 6.5 buffer + 0.05% SLS are represented by open symbols for the 100 μg/mL sample dose (◇), 500 μg/mL sample dose (△), 2000 μg/mL sample dose (○), and 10,000 μg/mL sample dose (□) (**b**). Results represent the average permeated triamcinolone concentration ± SD (*n* = 3).

**Figure 5 pharmaceutics-13-00323-f005:**
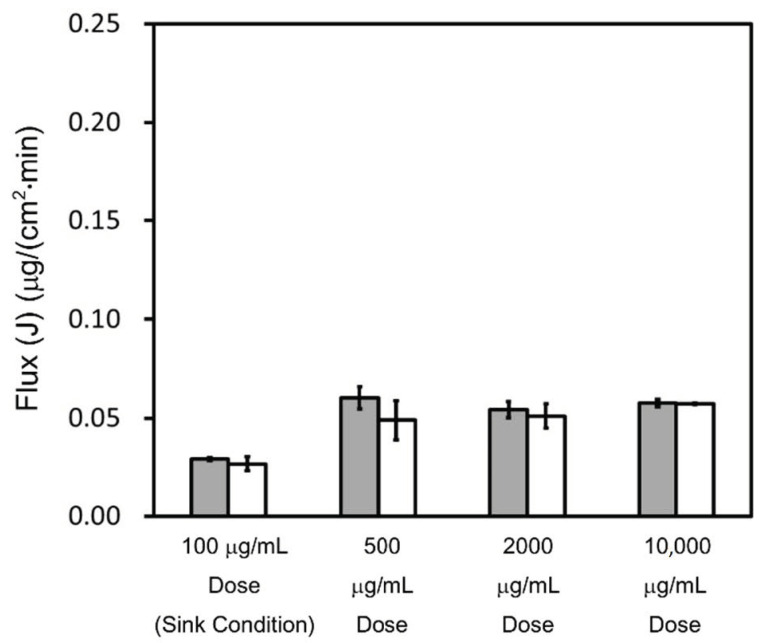
Calculated flux of triamcinolone. A time interval of 120–240 min was selected. The flux in pH 6.5 buffer is represented by gray bars. The flux in pH 6.5 buffer + 0.05% SLS is represented by white bars. Results represent the average triamcinolone flux ± SD (*n* = 3).

**Figure 6 pharmaceutics-13-00323-f006:**
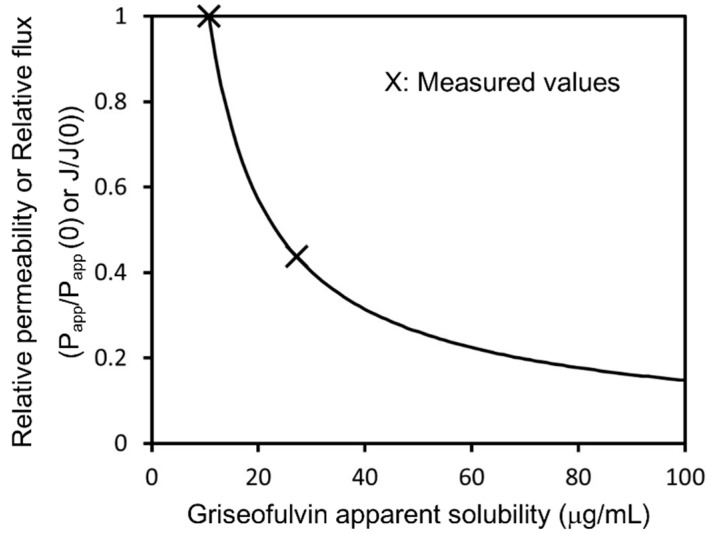
Calculated relationship between flux/apparent permeability and apparent solubility under sink conditions for griseofulvin. It was assumed that the sample concentration in the donor chamber remained at 5 μg/mL during the measurements. P_app_ (0) is the apparent permeability in the absence of solubilizer additives. J (0) is the flux in the absence of solubilizer additives.

**Figure 7 pharmaceutics-13-00323-f007:**
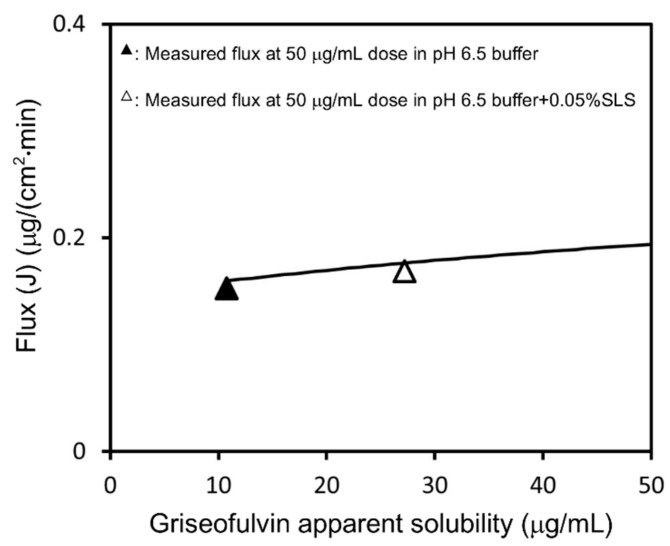
Calculated flux–apparent solubility relationship under sink conditions for griseofulvin. It is assumed that the UWL thickness remained constant. The results of flux measurement at a 50 μg/mL sample dose in pH 6.5 buffer and pH 6.5 buffer + 0.05% SLS were plotted.

**Figure 8 pharmaceutics-13-00323-f008:**
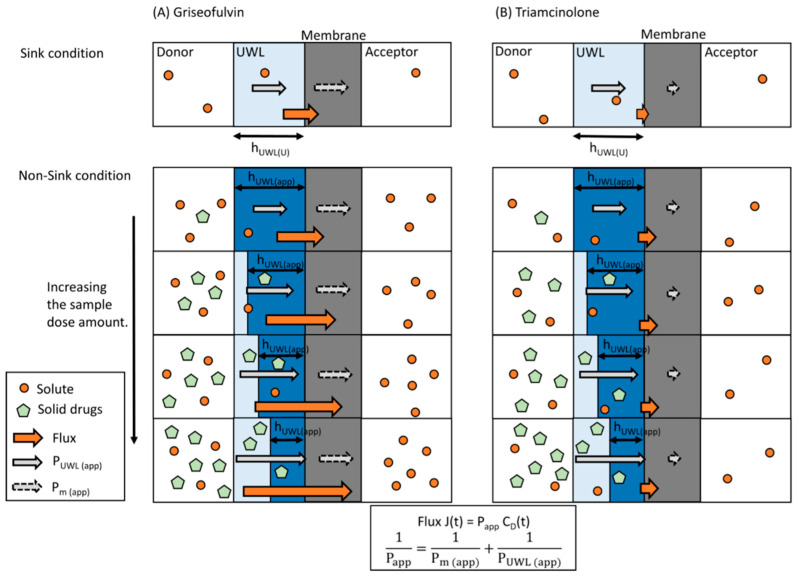
Proposed drug absorption mechanism under sink conditions and non-sink conditions for griseofulvin (**A**) and triamcinolone (**B**). Longer arrows represent higher flux, P_UWL (app)_, or P_m (app)_.

**Figure 9 pharmaceutics-13-00323-f009:**
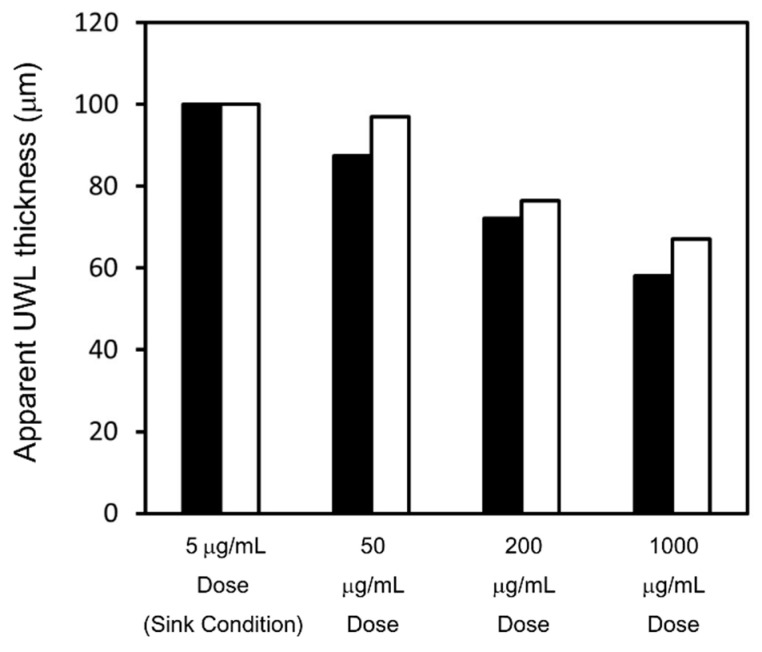
The calculated apparent UWL thickness (h_UWL (app)_) of griseofulvin for each flux measurement. h_UWL (app)_ in pH 6.5 buffer is represented by black bars. h_UWL (app)_ in pH 6.5 buffer + 0.05% SLS is represented by white bars.

**Figure 10 pharmaceutics-13-00323-f010:**
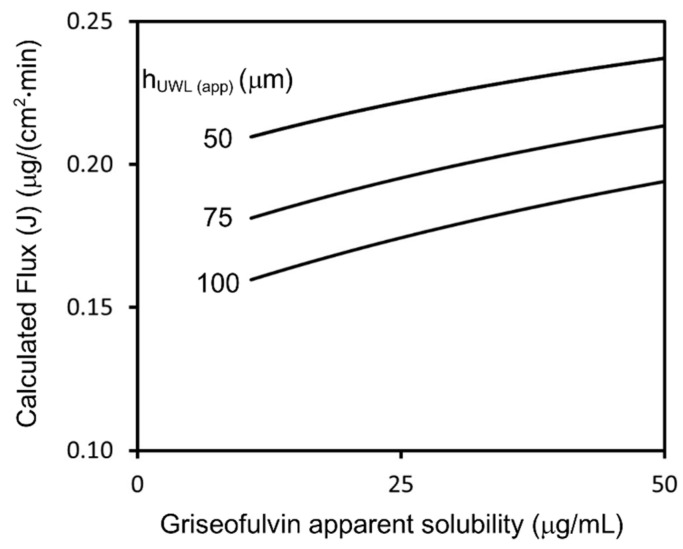
Calculated flux of griseofulvin for 50–100 μm apparent UWL thickness.

**Figure 11 pharmaceutics-13-00323-f011:**
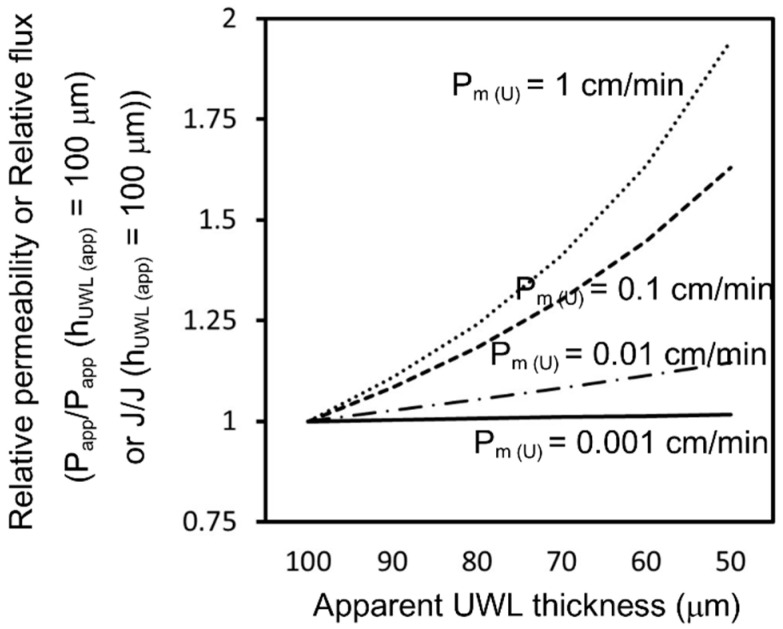
Calculated relationship between apparent permeability/flux and apparent UWL thickness for model compounds with P_m (U)_ of 0.001–1 cm/min. It was assumed that the MW of the model compounds was 400 Da.

**Table 1 pharmaceutics-13-00323-t001:** Physicochemical properties of griseofulvin and triamcinolone.

Physicochemical Properties	Griseofulvin	Triamcinolone
Molecular weight (MW)	352.77	394.43
Ionization properties	Neutral	Neutral
Log P ^1^	2.18 [40,41]	1.03 [40,42]
Aqueous solubility (μg/mL) ^2^	29.9 [40,41]	158 [40,42]

^1^ The partition coefficient is for partitioning between octanol and water. ^2^ Aqueous solubility was measured at 37 °C.

**Table 2 pharmaceutics-13-00323-t002:** List of test sample solution conditions in the donor chambers.

Compound	Test Media	Sample Dose Amount (μg/mL)
Griseofulvin	pH 6.5 phosphate buffer (pH 6.5 buffer)	5, 50, 200, and 1000
	pH 6.5 phosphate buffer with 0.05% (*w/w*) sodium lauryl sulfate (SLS) (pH 6.5 buffer + 0.05% SLS)	5, 50, 200, and 1000
Triamcinolone	pH 6.5 buffer	100, 500, 2000, and 10,000
	pH 6.5 buffer + 0.05% SLS	100, 500, 2000, and 10,000

**Table 3 pharmaceutics-13-00323-t003:** Solubility and un-ionized free drug (UFD) amount of griseofulvin and triamcinolone at 37 °C in each test medium.

Compound	Test Media	Solubility (μg/mL) ^1^	UFD Amount (μg/mL)	Fraction of UFD (F_U_)
Griseofulvin	pH 6.5 buffer	10.75 ± 0.38	10.75	1.00
	pH 6.5 buffer + 0.05% SLS	27.40 ± 0.07	10.75	0.39
Triamcinolone	pH 6.5 buffer	205.04 ± 10.34	205.04	1.00
	pH 6.5 buffer + 0.05% SLS	210.07 ± 6.54	205.04	0.98

^1^ Results represent average solubility ± Standard deviation (SD) (*n* = 3). UFD, un-ionized free drug.

**Table 4 pharmaceutics-13-00323-t004:** Measured and calculated permeability in pH 6.5 buffer.

Compound	Griseofulvin	Triamcinolone
MW	352.77	394.43
Measured P_app (U)_ (cm/min) ^1^	0.0148 ± 0.0007	0.000304 ± 0.000010
Calculated D_aq (U)_ (cm^2^/s)	5.18 × 10^−6^	4.93 × 10^−6^
Calculated P_UWL (U)_ (cm/min)	0.0311	0.0296
Calculated P_m (U)_ (cm/min)	0.0284	0.000307
Calculated P_app (U)_ (cm/min)	0.0148	0.000304

^1^ Results represent the average P_app (U)_ ± SD (*n* = 3). MW, molecular weight. UWL, unstirred water layer.

**Table 5 pharmaceutics-13-00323-t005:** Measured and calculated permeability in pH 6.5 buffer + 0.05% SLS.

Compound	Griseofulvin	Triamcinolone
Measured P_app_ (cm/min) ^1^	0.00648 ± 0.00026	0.000263 ± 0.000028
Calculated P_m (app)_ (cm/min)	0.0112	0.000300
Calculated P_UWL (app)_ (cm/min)	0.0153	0.0212
Calculated P_app_ (cm/min)	0.00648	0.000263
Calculated D_aq (app)_ (cm^2^/s)	2.56 × 10^−6^	3.54 × 10^−7^
Calculated D_aq (B)_ (cm^2^/s)	8.37 × 10^−7^	-

^1^ Results represent the average P_app_ ± SD (*n* = 3).

## Data Availability

The data presented in this study are available in this article.

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
