# Peer review of "Dose-Dependent Solubility–Permeability Interplay for Poorly Soluble Drugs under Non-Sink Conditions"

_pharmaceutics, 2021, doi:10.3390/pharmaceutics13030323_

Round 1

Reviewer 1 Report

This manuscript touches important solubility-permeability topic. Although this question is not new, the problems related to efficient drug delivery have direct connection to the solubility-permeability properties of formulations. The manuscript is well written (including methodologies), presentation is clear, and results are well discussed. Some comments:

  • Abstract has unexplained abbreviations

  • The manuscript would benefit many readers by escaping the monopoly discussing classical framework of formulation and connecting their work to the nano-formulations and drug delivery as well.

  • Can the changes in the flux between sink and non-sink conditions interpreted by altered the partitioning coefficient between different compartments? At least theoretically, the addition of SDS should change the phase preferences of dissolved molecules. There published studies that partitioning may be even more critical in drug release or permeation then diffusion coefficients. I wander what the authors think about it in in vitro and in vivo situations and there comments in discussion and introduction would be very useful.

  • In view of diffusion transport, the analysis might be rewarding if conducted in equimolar conditions because the permeability and flux is affected by concentration gradients.

  • Discussion could have a bit less analysis results; some of it could be part of Results. It is minor but could be helpful for readers.

Author Response

Thank you for your comments on our paper. They have helped us to improve the paper. We have answered each of your points. Please see the attachment.

Reviewer 2 Report

I found the abstract confusing and not cohesive. For instance, the gastrointestinal tract is mentioned at the end whereas nothing above or after it indicates why that is important. The largest structural issue with the manuscript is the separation of a Results and discussion Section. This seems unwise for this work and much of the discussion section would, in my opinion, be better placed in the Results section as they detail calculations and analysis of the experimental data. Overall, the data and experiments seem good, but the presentation should be more streamlined. The manuscript should also be edited thoroughly by a native English speaker as there were numerous grammatical issues throughout.   Other comments (Lines given): 65-85) The introduction includes a discussion on sink and non-sink conditions without clearly defining or explaining what that means. 185) I found much of the method section somewhat unclear as to exactly what was done and why. For instance, the authors state “The solubility of each model compound at 37°C for each test media was determined 185 by using the donor chamber sample concentration at 0 min at the highest sample dose.” I don't understand the approach. The donor chamber concentrations are set by the authors, so I don’t understand how this is a measurement of the compound solubility. 202) Looking at Table 2, I can imaging how the solubility study was done. But the text is unclear. Perhaps it would be better to separate the measurement of solubility from the performance of the flux measurements. These are really two different measurements and should be treated as such in the methods (they are in the Results). 224) I don’t understand the reasoning behind the assumption that the UFD amount in Table 3 would be equal to the solubility in pH6.5buffer both with and without SLS. Isn’t it also possible for there to be additional UFD present due to differences in the solution properties and yet not bound to SLS. This type of phenomena is widespread (as the authors know) for the solubility of many compounds in solvent mixtures and solvents with solubilizing additives. 226-236) Section 4.2 and Table 4 have very limited description of the results. Perhaps the manuscript is unclear, but at this point in the manuscript I don’t understand the distinctions in the Table. The nominal sample dose amount is the actual dose put in the donor chamber, sure. The actual sample dose amount is then, what? The actual solution concentration? How then can they be above the solubility? Or is this a report of the inability of the experimentalist to hit their target concentrations? Next, the solid drug concentration, is “calculated by subtracting the sample concentration at 0 min from the sample dose amount”. What? Wouldn’t the concentration at 0 min be either the nominal dose or the solubility? Second, no other data presented here includes temporal information. Perhaps data is omitted? No, it comes later. Overall, confusingly presented or just lacking clarity. 248-251) Right, you have saturated solutions in equilibrium with the solid as would be expected. 257-262) There seems to be a lot that is being overlooked here as both the donor and acceptor cell concentrations are transient in experiments such as these. While they both begin at approximately the same concentration in the donor cell, one might expect there to be differing equilibration kinetics with respect to the dissolution of the drug as some diffuses to the acceptor cell. Additionally, the water flux from the acceptor cell to the donor cell is also neglected (which can sometimes be valid, sometimes not) which would also lead to differing dissolution kinetics. More critically, the use of Equation 12 and the generalized way this is discussed seems unnecessary. As you have the acceptor cell concentrations you one would normally conduct diffusion cell experiments such as these to measure the solute permeabilities directly. 269) The higher flux in the 50, 200, 1000 would be expected from the higher driving force for transport. However, if the additional drug is complexed with the SLS, as the authors assume earlier in the manuscript, it’s flux would be dependent on the diffusion of that larger complex and thereby slower. So, is this a difference in the UFD permeability or just an experimental artifact of simultaneously measuring both UFD and complexed drug permeation? 273-305) similar comments as above on the analysis and lack of analysis of this drug system. Section 5 Discussion) The separation of many of the things in the Discussion from the Results section is unwise. Indeed, many things in the discussion are actually results. For instance, the Permeability from the experiments! By this point in the paper, you would have lost this reader’s interest. 390) Figure 8 is a good one. It should come much earlier in the manuscript as it gives the results context. 468) A CRediT statement would be easier to parse.

Author Response

(The authors gave the same response as above.)

Reviewer 3 Report

The text bellow contains comments on manuscript entitled “Dose-dependent solubility-permeability interplay for poorly soluble drugs under non-sink conditions” given for revision.

The manuscript is focused on the investigation of the non-sink conditions and SLS, on the solubility and permeability of two poorly soluble drugs (griseofulvin and triamcinolone) at various doses, with or without SLS, by flux measurements. It was found that the drug molecules bound to solubilizer additives and solid drugs could enhance UWL permeability, resulting in an increased flux or permeability under non-sink conditions. By combining flux measurements with physiologically based pharma-cokinetics (PBPK) modeling, it would be able to accurately predict the in vivo absorption for BCS Class II and IV drugs in the future.

To my opinion the manuscript merit to be published. It well organized with logically structured experimental design. The results and discussions fully explain the goal of the study.

I would highly recommend the authors to read and correct the English language in terms of grammar and language style. I have underlined some of the most common, but the authors also must pay attention.

Page 1, Line 14. Please correct to ... griseofulvin, …

Page 1, Line 44. Is it necessary (pp. 297-307) to stay there?

Page 2, Line 50. I think it will be better if it is written: Drug molecules dissolved …

Page 2, Line 90: Please change to: We used griseofulvin and triamcinolone …

Please use small caps for griseofulvin and triamcinolone when they are in the middle of the sentence

Page 5, Line 154. Please correct to: Twenty mL …

Please do not start the sentence with a number or abbreviation.

Author Response

(The authors gave the same response as above.)

Round 2

Reviewer 2 Report

the authors did a fairly good job updating the manuscript to address the previous comments.